# Urinary cotinine concentration as a biomarker of environmental exposure to Nicotine in Vietnam: Results from a Nationwide Survey in 2024

Hai Thanh Le[1], Thuy Thi Thu Tran[2]*, Binh Thi Ta[3], Huyen Thi Nguyen[3], Van Thi Pham[3], Chinh Thuy Thi Phan[4], Son Duc Nguyen[5], Nhat Minh Tu[6], Quynh Thuy Nguyen[2], Bich Ngoc Nguyen[2], Son Van Nguyen[1]

1 National Institute of Occupational and Environmental Health, Ministry of Health, Hanoi, Vietnam, 2 Faculty of Environmental and Occupational Health, Hanoi University of Public Health, Hanoi, Vietnam, 3 Department of Medical Testing and Environmental Analysis, National Institute of Occupational and Environmental Health, Ministry of Health, Hanoi, Vietnam, 4 The Center for Training and Scientific Management, National Institute of Occupational and Environmental Health, Ministry of Health, Hanoi, Vietnam, 5 National Institute of Medical Device and Construction, Ministry of Health, Hanoi, Vietnam, 6 Department of Biotechnology (Drug Development), University of Science and Technology of Hanoi, Vietnam Academy of Science and Technology, Hanoi, Vietnam

* tttt@huph.edu.vn

## Abstract

### Background

Accurate sources on environmental nicotine exposure, such as biomarker data, remain insufficient in low- and middle-income countries. This study aimed to i) determine the optimal cut-off point of urinary cotinine that discriminates smokers from non-smokers, ii) estimate misclassification rate between self-reported smoking and urinary cotinine, and III) explore the distribution of tobacco smoke exposure levels using urinary cotinine concentrations among adults in Vietnam in 2024.

### Methods

A cross-sectional study was conducted in 2024 across seven provinces representing Vietnam's ecological regions. Using multi-level stratified random sampling techniques, 1,077 adults aged 18–60 were recruited. Demographic and behavioural data were obtained through structured interviews. Urinary cotinine to creatinine ratios (CCR) were measured using high-performance liquid chromatography-tandem mass spectrometry. The Youden J method was used to determine the optimal cut-off point of CCR. Statistical analyses were performed using SPSS 20.0.

### Results

Self-reported results showed that 18.3% were active smokers, 33.4% were exposed to SHS at home, and 48.3% lived in a non-smoking household. The optimal CCR cut-off value of 20.947 μg/g can distinguish smokers and non-smokers with a sensitivity

**Data availability statement:** All relevant data are within the paper and its Supporting information files.

**Funding:** SVN received the funding from the Ministry of Science and Technology as a principle investigator of the National Project "Development of a Basic Biochemical Index Set for Environmental Toxicology in the Vietnamese Working-Age Population" (Project code: ĐTĐL.CN-39/21) under the Basic Science Development Program in Chemistry, Life Sciences, Earth Sciences, and Marine Sciences (2017–2025). The funders had no role in study design, data collection and analysis, decision to publish, or preparation of the manuscript.

**Competing interests:** The authors have declared that no competing interests exist.

of 61.5%, specificity of 93.2%, 70.6% positive predictive value and 90% negative predictive value. Regional disparities and urinary cotinine among the non-smoking groups suggest potential environmental exposure to nicotine.

## Conclusion

The CCR level of 20.947 µg/g indicated the optimal cut-off value to distinguish smokers and non-smokers. Vietnam was among countries with high levels of environmental nicotine exposure, with significant variation by sex, education, occupation, income, and region. Urinary cotinine is a reliable biomarker for nicotine exposure and should be integrated into routine surveillance. These findings support the need for stricter enforcement of smoke-free environments and interventions tailored to reduce involuntary tobacco exposure.

## Introduction

Nicotine is the primary addictive component of tobacco. Studies have demonstrated the health impact of nicotine on both active smokers and individuals exposed to tobacco smoke in the environment [1]. Environmental exposure to nicotine occurs via passive smoking, such as second-hand smoke (SHS) and thirdhand smoke (THS), which have been linked to respiratory diseases, cardiovascular dysfunction, impaired neurodevelopment, and other adverse outcomes, particularly among vulnerable populations such as children, pregnant women and people with chronic diseases [2–7].

Both active and passive smokers are exposed to more than 7,000 chemicals in the tobacco smoke, hundreds of which are toxic, and at least 69 are reported carcinogens [1]. A global report of the World Health Organisation reveals that about one-third of the population have frequent SHS exposure, contributing to approximately 1.3 million deaths and 37 million disability-adjusted life-years annually among non-smokers [8]. Besides SHS, THS has become a major problem because residual nicotine and other chemicals persist on surfaces and respirable particles long after tobacco smoke has been emitted for months or even years. These residues can undergo chemical reactions with common indoor pollutants, such as ozone and nitrous acid, forming secondary toxicants, including carcinogenic nitrosamines [9,10]. Thus, even smoke-free environments may remain hazardous for prolonged periods [9,10]. While the direct impact of tobacco smoke among active smokers is available in a body of literature, accurate evidence of SHS and THS exposures remains limited. Therefore, monitoring environmental nicotine exposure is crucial for understanding the extent of SHS and THS contamination and informing effective public health interventions.

Environmental monitoring methods vary, including air or surface samplings. Passive air nicotine monitors can measure cumulative ambient nicotine concentrations over extended periods [11]. Surface wipe sampling and dust analysis complement air monitoring by detecting residual nicotine accumulation on surfaces, offering insights into cumulative exposure and long-term THS-related risks [9,11]. However, environmental measurements may not fully reflect individual-level exposure variations, influenced by

time-activity patterns and personal behaviours. Therefore, biological monitoring, particularly the measurement of cotinine—the major metabolite of nicotine—in biological fluids such as urine, saliva, or serum, is considered the gold standard for assessing exposure [4,12]. Cotinine's half-life of approximately 16 hours allows detection of recent nicotine exposure within a 2–4-day window [12]. Serum cotinine measurement, while highly sensitive, requires invasive blood collection, and salivary cotinine may be less stable in some settings [12]. Urinary cotinine measurement is favoured in large-scale studies due to its non-invasive nature, high sensitivity, and ability to detect recent exposures through multiple routes [4,12].

Despite the availability of robust methods, important gaps persist, particularly in low- and middle-income countries (LMICs) where tobacco use prevalence remains high, but systematic environmental or biological surveillance of environmental nicotine exposure is limited [2]. Research from high-income countries, such as the United States and parts of Europe, consistently shows that a substantial proportion of non-smokers exhibit measurable cotinine levels [13–16]. Data from the U.S. National Health and Nutrition Examination Survey (NHANES) 2017–2020, for example, found that approximately 40% of non-smoking adults had detectable urinary cotinine [13]. However, less systematic data are available from LMICs, where enforcement of smoke-free policies may be inconsistent and social norms surrounding smoking differ. Studies conducted in public venues in several Latin American countries found air nicotine concentrations exceeding WHO-recommended limits by 2–5 times where smoking bans were poorly enforced; similar findings were found in Gambia in 2021 [17,18]. In addition, studies have reported the unintended consequences of smoke-free policy on the increase of indoor exposure to nicotine at home via SHS and THS, especially among vulnerable populations such as children and pregnant women [19–22].

Vietnam has made significant progress in tobacco control over the past decade with the enforcement of the Law on Prevention and Control of Tobacco Harm in 2012 and recent updates like Circular No. 11/2023/TT-BYT promoting smoke-free environments. Although smoking prevalence among adult men declined from 47.4% in 2010 to 45.3% in 2015 [23], exposure to SHS remains widespread. The 2022 Vietnam Population-Based Provincial Global Adult Tobacco Survey (PGATS) reported that 44.4% of adults aged 15 and older were exposed to SHS at home, and 23.1% at their workplace [2]. Notably, women were disproportionately exposed at home, underscoring sex-based vulnerabilities [24].

Vietnam's current tobacco surveillance efforts primarily focus on self-reported tobacco exposure, which can be affected by recall bias and social desirability [14,15], particularly in cultural contexts where male smoking is normalized and smoking among women and children is stigmatized [2,23]. Furthermore, research and policy frameworks in Vietnam have largely overlooked THS as a distinct source of environmental contamination and exposure risk.

Given the substantial health risks associated with SHS and THS exposure, the persistence of nicotine contamination in indoor environments, and the current lack of objective biomarker-based exposure assessments in Vietnam, there is a pressing need for comprehensive and systematic monitoring using validated environmental and biological monitoring techniques. This first nationally representative biomarker survey aims to i) determine the optimal cut-off point of urine cotinine that discriminates smokers from non-smokers, ii) estimate misclassification rate between self-reported smoking and urinary cotinine, and III) explore the distribution of tobacco smoke exposure levels using urinary cotinine concentrations among adults in Vietnam in 2024. This study was a part of the National Project titled "Development of a Basic Biochemical Index Set for Environmental Toxicology in the Vietnamese Working-Age Population" in 2025 under the management of the National Institute of Occupational and Environmental Health.

## Materials and methods

### Study design and population

This cross-sectional study was conducted from April to August 2024 across seven Vietnamese provinces representing different geographical regions: Lang Son (Northeastern), Hoa Binh (Northwestern), Ha Noi (Red River Delta), Nghe An (North Central), Khanh Hoa (South Central and Central Highlands), Ho Chi Minh City (Southeastern), and Can Tho (Mekong Delta).

A total of 1,077 adults aged 18–60 years were recruited. Eligible participants were healthy at the time of sampling, without chronic diseases requiring treatment (e.g., hypertension, diabetes, chronic obstructive pulmonary disease, metabolic disorders, or cancer). Exclusion criteria included pregnancy, performing strenuous physical activity, direct exposure to chemicals at work, or living near industrial pollution areas.

Study participants were selected via multi-stage sampling procedure as follow:

Stage 1. Selecting provinces: To ensure that the research sample represents the Vietnamese working-age population in different parts of Vietnam, we randomly selected one province from each of seven administrative and socio-ecological regions of the country. The selected provinces included Lang Son province (Northeast region), Phu Tho province (old Hoa Binh, Northwest region), Hanoi city (Red River Delta region), Nghe An province (North Central region), Khanh Hoa province (South Central Coast and Central Highlands region). Ho Chi Minh city (Southeast region), Can Tho city (Mekong Delta region).

Stage 2. Selecting commune/ward areas: In each province/city, we randomly selected one district/county. In each district/county, we intentionally selected one ward/commune that was not located near mining, metallurgy, pesticide production, large industrial zones, had no craft villages with the risk of exposure to toxic chemicals, and no cultivation of industrial crops. Seven ward/communes were invited to join the study.

Phase 3. Selecting subjects. Local health workers under the guidance of study staff prepared a list of subjects who met the study inclusion criteria. Subjects were stratified by sex (approximately 1:1 male-to-female ratio) and across four age groups (18–24, 25–34, 35–44, 45–60 years). From the above list, 20 subjects were randomly selected in each age group with equal sex ratio. Thus, 160 subjects (80 men and 80 women) were invited to join the study in each commune/ward. During the data collection process, if the subject refused to join or end their participation, another study subject would be randomly selected from the initial list. In total, 1077 healthy Vietnamese people in working age participated in the study.

## Data/ Sample collection

Data were collected through structured, face-to-face interviews by trained staff at the time of urine sample collection. The structured questionnaire was developed specifically to gather information on participants' characteristics. A pilot test was conducted with a small sample of participants to evaluate the comprehensibility, estimated completion time, and logical flow of the questions. Revisions were made based on pilot findings, including adjustments to wording, phrasing, and item sequencing. The finalised version of the questionnaire was used for formal data collection.

The study staff sought for approval from the Commune People's Committee for the study implementation. Trained local healthcare workers served as interviewers for the study. Invitation letters were sent to participants in advance of scheduled sessions at the commune or ward health station. Upon arrival, participants were provided with information about the study's aims and procedures. Written informed consent was obtained before participation. Five data collectors conducted face-to-face interviews, each lasting approximately 20 minutes. Completed questionnaires were collected immediately upon conclusion of each interview.

Urine samples were collected using standardised self-collection kits after interviews at study sites. Participants provided midstream urine samples to minimise contamination. Approximately 3–4 mL was collected for creatinine analysis, and 45 mL for cotinine analysis. Samples were transported at 4–8°C and stored at −20°C until analysis. Participant recruitment and data/sample collection were conducted concurrently from 21/05/2024 to 05/07/2024.

## Measures

The questionnaire covered demographic information, lifestyle behaviours, including smoking and tobacco smoke exposure at home. Demographic data included age (18–24; 25–34; 35–44 and 45–60 years), sex (male or female), marital

status (married and others), education level (elementary, junior high school, high school and college and above), occupation (homemakers, farmers, workers, officials and others professions), and monthly income (< 1 million, 1–5 million, 5–10 million, > 10 million VND). Lifestyle habits were assessed, including work-related travel, which was categorised into three levels: rarely, occasionally, and frequently. Alcohol consumption and physical exercise habits were recorded as binary responses (yes or no).

Exposure to tobacco smoke was recorded through three variables: Active smoking, Second-hand smoke exposure at home and Non-smoking exposure at home. Active smoking was defined as currently smoking either cigarettes or pipe tobacco. Second-hand smoke exposure at home was characterised by a non-smoking individual's exposure to second-hand tobacco smoke resulting from the presence of smokers within the household. No smoking at home included individuals who reported no smoking and living in households free of smokers.

The full questionnaire is available in S1 File.

## Laboratory analysis

Urine analysis was conducted at the Department of Medical Testing and Environmental Analysis, National Institute of Occupational and Environmental Health (NIOEH), accredited under ISO/IEC 17025:2017 including nicotine/cotinine technique. Cotinine concentrations were quantified using high-performance liquid chromatography (Agilent 1290 Infinity HPLC) coupled with tandem mass spectrometry (QTRAP 3200, SCIEX), following a validated method [25].

Sample preparation involved solid-phase extraction using HLB columns (60 mg/3 mL). Columns were preconditioned with methanol and deionised water, and 1 mL of urine was loaded. Analytes were eluted with 1mL of 10% formic acid in methanol and 10% formic acid (1:1). Certified reference materials (ClinChek® Urine Controls, Germany) ensured quality control. The limit of detection (LOD) was 0.03 μg/L for cotinine, with a mean recovery rate of 101.9%.

Certified reference materials (CRM), ClinChek® Urine Controls, Germany level I, were used for quality control the results of LC MS/MS analysis. Measured results and recoveries for cotinine in CRM ClinChek® Urine Controls, Germany were shown in the Table 1. CRM had a target value of 163 μg/L, the allowable value range was from 130 μg/L to 196 μg/L.

In epidemiological research, spot urine sampling is frequently preferred due to its methodological advantages: it involves the collection of a single specimen, making it logistically simple and minimally disruptive to participants' daily routines, while also reducing material and operational costs. Nonetheless, spot urine samples are susceptible to variability in analyte concentration resulting from fluctuations in urine dilution. Therefore, biomarker concentrations should be adjusted for urinary creatinine concentrations. The normalisation of analyte concentrations to urinary creatinine levels is a widely adopted approach to correct for variations in urine dilution, thereby enhancing the analytical accuracy and inter-sample comparability [26,27]. In this study, a cotinine-to-creatinine ratio (CCR) was used as a biomarker of tobacco smoke exposure.

Table 1. Measured results and recoveries for cotinine in Certified reference materials ClinChek® Urine Controls.

| No | Sample | Cotinine concentration (μg/L) | Recovery (%) | Average cotinine concentration (μg/L) | Standard deviation (SD) | Relative standard deviation (RSD%) |
|---|---|---|---|---|---|---|
| 1 | CRM$_1$ | 162.69 | 99.81 | 166.16 | 7.74 | 4.66 |
| 2 | CRM$_2$ | 165.39 | 101.47 | | | |
| 3 | CRM$_3$ | 173.47 | 106.43 | | | |
| 4 | CRM$_4$ | 170.78 | 104.77 | | | |
| 5 | CRM$_5$ | 176.17 | 108.08 | | | |
| 6 | CRM$_6$ | 160.00 | 98.16 | | | |
| 7 | CRM$_7$ | 154.61 | 94.85 | | | |

## Statistical analysis

Statistical analyses were conducted using SPSS version 20.0 (IBM Corp., USA). Cotinine concentrations below the limit of detection (LOD) were imputed as LOD divided by 5. Descriptive statistics included the number of observations (N) and corresponding percentages (%). Given that the distribution of the cotinine-creatinine ratio (CCR) was non-normal, the median and interquartile range (IQR) were reported, and group differences were assessed using the Mann-Whitney U test and Kruskal–Wallis test.

Cotinine concentrations normalized by urinary creatinine were evaluated for extreme values using multiple statistical approaches, including the interquartile range (IQR) rule, z-scores ($> |3|$), and the 99th percentile threshold. A biological plausibility check was also performed based on published reference ranges for urinary cotinine in smokers. Data points that were identified as extreme by at least two independent methods were classified as outliers. Using this combined approach, 32 outliers (2.97% of total samples) were flagged. These values were retained in the dataset with an indicator variable for sensitivity analyses and model robustness checks.

To identify the optimal CCR cut-off for distinguishing smokers from non-smokers, Receiver Operating Characteristic (ROC) analysis was employed, following the method introduced by Youden (1950) and widely adopted in subsequent research [28–30]. This approach identifies the cut-off point that maximises the difference between the true positive rate and false positive rate, aiming to classify individuals as accurately as possible. The ROC curve plots sensitivity against 1-specificity for all potential cut-off values of CCR.

A CCR cut-off point yielding an area under the ROC curve (AUC) between 0.8 and 1.0 indicates excellent discrimination of active smokers. An AUC between 0.7 and 0.8 reflects moderate accuracy, while values below 0.7 suggest poor discriminatory ability. An AUC of 0.5 or lower implies no meaningful distinction between smokers and non-smokers based on urinary CCR [30]. The distribution of tobacco smoking status based on the optimal CCR cut-off point across participant characteristics was evaluated using Chi-square tests, odd ratios and 95% confidence intervals (CI). Statistical significance was set at $p < 0.05$.

## Ethics statement

The study protocol was approved by the Ethics Committee in Biomedical Research at the National Institute of Occupational and Environmental Health, Ministry of Health, Vietnam (Certificate No. 01/GCN-HDDD, issued March 29, 2024). All participants provided a written informed consent before enrolment. Personal information was kept confidential and accessed only by the main investigator for this study's purpose.

## Results

With the stratification sampling method, the number of participants was equal across provinces, age groups and sexes (Table 2). More than half of the participants were married (65.9%) and finished high school or higher education (60.5%). More participants worked in informal sectors (73.9%) with a monthly income of over 5 million VND (54.7%).

Among the 1,077 study participants, more than half were either active smokers (18.3%) or SHS at home (33.4%), and about 48.3% reported no smoking exposure in their family.

Difference in CCR was observed among groups of province, sex, education, occupation, work-related travel and alcohol consumption. Among provinces, Ho Chi Minh City had the highest median of CCR, followed by Can Tho and Lang Son; Hanoi City had the lowest median. The median of CCR among males was five times higher than that among female participants. Negative correlation between CCR and education levels was observed, with lower educational levels reporting higher CCR. Participants with other jobs (freelancers, small business owners, etc) had a higher CCR level than participants with an office job. More than 80% participants had to travel because of work to some extent, and they also had higher CCR than those who did not have to travel for work. Participants with alcohol consumption also showed higher levels of CCR.

**Table 2.  Characteristics of study participants and Cotinine to creatinine ratio (CCR) (µg/g creatinine) (N = 1077).**

| Variables | Sub-groups | Total | CCR (µg/g creatinine) | Nonparametric tests[a] |
|---|---|---|---|---|
| | | n (%) | Median (±IQR) | p |
| Province categories | Lang Son | 160 (14.9) | 0.9 (±12.3) | <0.01 |
| | Hoa Binh | 155 (14.4) | 0.4 (±4.1) | 1 |
| | Ha Noi | 156 (14.5) | 0.01 (±3.7) | Ref |
| | Nghe An | 149 (13.8) | 0.8 (±7.6) | 0.12 |
| | Ho Chi Minh | 157 (14.6) | 2.8 (±49.8) | <0.01 |
| | Can Tho | 152 (14.1) | 2.2 (±10.3) | <0.01 |
| | Khanh Hoa | 148 (13.7) | 0.3 (±11.2) | 1 |
| Age categories | 18-24 years | 274 (25.4) | 1.1 (±31.7) | 0.21 |
| | 25-34 years | 266 (24.7) | 0.7 (±5.7) | |
| | 35-44 years | 270 (25.1) | 1 (±14.2) | |
| | 45-60 years | 267 (24.8) | 0.8 (±9.3) | |
| Sex | Male | 511 (47.4) | 2.5 (±119.7) | <0.01 |
| | Female | 566 (52.6) | 0.5 (±2.8) | Ref |
| Marriage status | Married | 710 (65.9) | 0.8 (±7.6) | Ref |
| | Single/widow/ divorced | 367 (34.1) | 1.1 (±21.1) | 0.20 |
| Education | Elementary | 166 (15.4) | 1.4 (±13.9) | <0.01 |
| | Junior high school | 259 (24) | 1.4 (±47.3) | 0.03 |
| | High school | 365 (33.9) | 0.9 (±10.3) | 0.13 |
| | College and higher | 287 (26.6) | 0.5 (±4.9) | Ref |
| Occupation | Farmers | 224 (20.8) | 0.6 (±5.6) | 1 |
| | Workers | 93 (8.6) | 0.9 (±11.8) | 0.09 |
| | Officials | 188 (17.5) | 0.5 (±4.2) | Ref |
| | Others | 572 (53.1) | 1.4 (±19.1) | <0.01 |
| Income per month | < 1 million | 221 (20.5) | 1 (±9.6) | 0.86 |
| | 1 - 5 million | 267 (24.8) | 0.9 (±7.3) | |
| | 5-10 million | 508 (47.2) | 0.9 (±9.8) | |
| | >10 million | 81 (7.5) | 0.7 (±130.8) | |
| Work requires frequent travel | No | 184 (17.1) | 0.5 (±3.3) | Ref |
| | Sometimes | 246 (22.8) | 1.1 (±8.5) | 0.03 |
| | Often | 647 (60.1) | 1.1 (±16.7) | 0.02 |
| Drink alcohol | Yes | 361 (33.5) | 2.5 (±122.4) | <0.01 |
| | No | 716 (66.5) | 0.6 (±4.7) | Ref |
| Exercise | No | 599 (55.6) | 0.9 (±8.6) | 0.87 |
| | Yes | 478 (44.4) | 0.9 (±13.4) | |
| Tobacco smoke exposure | Active smoking | 197 (18.3) | 131.7 (±335.9) | <0.01 |
| | Second-hand smoke exposure at home | 360 (33.4) | 0.8 (±8.9) | <0.01 |
| | No smoking exposure at home | 520 (48.3) | 0.5 (±2.2) | Ref |

[a]Mann-Whitney test for 2 samples, Kruskal-Wallis for more than 2 samples.

The CCR was appropriately correlated with the self-reported status of tobacco smoke exposure, in which the median of CCR among active smoking and SHS groups was significantly higher than that of the No household smoking exposure group.

The cut-off point for CCR at 20.947 µg/g creatinine gave a good classification between smokers and non-smokers as shown by the Receiver Operating Characteristic (ROC) curve with area under the curve, AUC = 0.83 (sensitivity: 0.706; specificity: 0.901) (Fig 1).

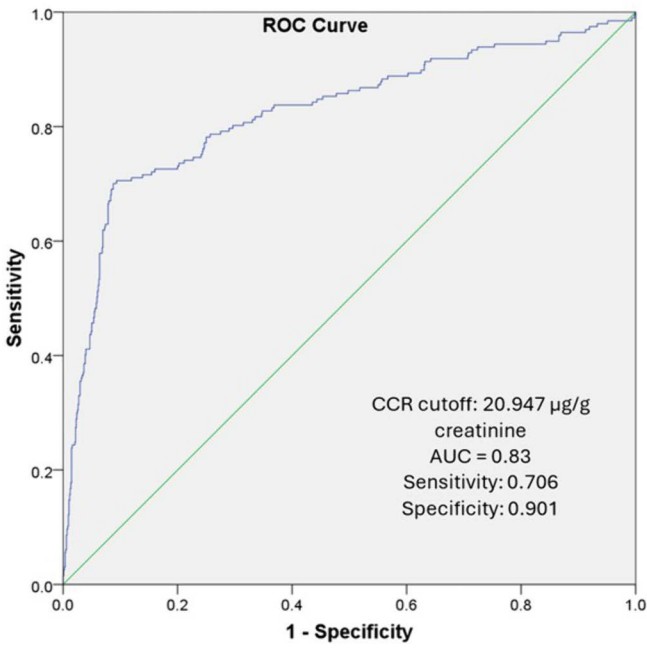

**Fig 1. Receiver Operating Curve Characteristics (ROC) of the CCR.** Smokers were well differentiated from non-smokers; AUC: 0.83.

The majority of the smokers had urinary CCR more than 20.947 µg/g creatinine (70.6%), while 80.8% SHS participants and 96.5% participants without tobacco exposure at home had CCR levels less than 20.947 µg/g creatinine (Fig 2).

The identified cut-off value demonstrated good diagnostic performance, with a sensitivity of 61.5% (true positive rate) and a specificity of 93.2% (true negative rate). The positive predictive value was 70.6%, while the negative predictive value reached 90.1%. Correspondingly, the false positive and false negative rates were relatively low, at 9.9% and 29.4%, respectively (Table 3). Notably, the prevalence of active smoking estimated by self-reported questionnaire was 18.3%, which was lower than the 21.0% prevalence determined using a CCR threshold above 20.947 µg/g creatinine.

Participants from Ho Chi Minh City and Khanh Hoa Province were more likely to be active smokers compared to those residing in Hanoi, with odds ratios (OR) of 2.4 (95% CI: 1.4–4.2) and 1.7 (95% CI: 1.0–3.1), respectively. Male participants

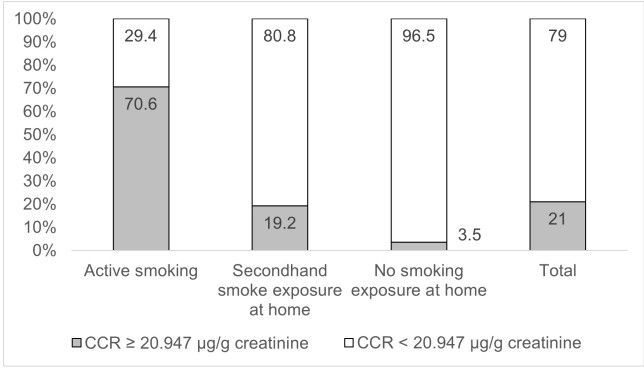

**Fig 2. Distribution of smoking exposure statuses by CCR cut-off value of 20.947 µg/g.**

**Table 3.** Diagnostic parameters of cotinine concentration.

| Self-reported smoking exposure | CCR | | | | Total | | |
|---|---|---|---|---|---|---|---|
| | Smokers | | Non-Smokers | | | | |
| | n | % | n | % | n | % | |
| Smokers | 139 (TP) | 70.6 | 58 (FN) | 29.4 | 197 | 18.3 | PPV: 0.706 |
| Non-smokers | 87 (FP) | 9.9 | 793 (TN) | 90.1 | 880 | 81.7 | NPV: 0.9 |
| Total | 226 | 21.0 | 851 | 79.0 | 1077 | 100.0 | |
| | Sensitivity: 0.615 | | Specifically: 0.932 | | | | |

Note: TP: True positive, FP: False positive, TN: True negative, FN: False negative, PPV: Positive predictive value, NPV: Negative predictive value.

had significantly higher odds of being active smokers than females (OR = 6.6, 95% CI: 4.5–9.4). Marital status appeared to be a protective factor, as married individuals had reduced odds of smoking (OR = 0.7, 95% CI: 0.5–0.9). Smoking prevalence was notably higher among those with lower levels of education compared to individuals holding college degrees or higher qualifications.

Additionally, participants with monthly incomes exceeding 10 million VND exhibited higher smoking prevalence than those earning between 1–5 million or 5–10 million VND. Individuals who frequently travelled for work had nearly twice the odds of being smokers compared to those who did not (OR = 1.9, 95% CI: 1.2–3.0). Smoking was also strongly associated with alcohol use, with drinkers being three times more likely to be active smokers than non-drinkers (OR = 3.0, 95% CI: 2.2–4.1) (Table 4).

## Discussion

This nationwide biomonitoring study provides the first comprehensive data on urinary cotinine levels and associated factors of tobacco smoke exposure among the Vietnamese adult population. The findings confirm that environmental nicotine exposure—both from active smoking and second-hand smoke (SHS)—remains widespread and socially patterned, despite significant policy advances in tobacco control in Vietnam. These results underscore the need for more targeted, inclusive public health interventions that go beyond self-reported exposure to incorporate biological monitoring and address emerging risks like thirdhand smoke (THS).

Nearly half of the population (48.3%) reported that they were not exposed to tobacco smoke in their household environment, but 18.3% were active smokers and 33.4% were exposed to SHS at home. These findings are consistent with previous national reports that highlight high rates of tobacco use and SHS exposure in Vietnam, particularly among men and in rural areas [23]. The 2022 Vietnam PGATS survey documented that approximately 44.4% of adults were exposed to SHS at home and 23.1% at work [2]. Compared with the 2015 Global Adult Tobacco Survey, where 22.5% of adults in Vietnam reported being smokers and nearly 60% reported SHS exposure at home [23], our findings suggest a slight decline in smoking rates but a persistent burden of SHS exposure.

Vietnam has made significant progress in implementing smoke-free policies as part of its broader tobacco control strategy under the Framework Convention on Tobacco Control (FCTC), which the country ratified in 2004. The primary legal instrument governing tobacco control is the Law on Prevention and Control of Tobacco Harms (Law No. 09/2012/QH13), which came into effect in 2013. This law mandates smoke-free environments in a wide range of public and indoor spaces, including health facilities, educational institutions, workplaces, and public transport [31].

Enforcement mechanisms include inspections and administrative penalties for violations are in place, however, implementation remains inconsistent in practice, especially in hospitality venues and some workplaces such as hotels and restaurants [32,33]. This may explain the detectable of CCR among groups of SHS and no smoking exposure at home. Continued efforts are needed to strengthen enforcement, raise public awareness, and integrate smoke-free policy compliance into broader health promotion strategies.

**Table 4. Distribution of the smoking status based on the optimal cut-off point of CCR (N = 1077).**

| Variables | Sub-groups | Total | Smokers (CCR ≥ 20.947 µg/g creatinine) | Non-smokers (CCR < 20.947 µg/g creatinine) | Adjusted OR | 95% Confidence Interval | |
|---|---|---|---|---|---|---|---|
| | | n | n (%) | n (%) | | Lower | Upper |
| Province categories | Lang Son | 160 | 33 (20.6) | 127 (79.4) | 1.5 | 0.8 | 2.7 |
| | Hoa Binh | 155 | 27 (17.4) | 128 (82.6) | 1.2 | 0.7 | 2.2 |
| | Ha Noi | 156 | 23 (14.7) | 133 (85.3) | ref | | |
| | Nghe An | 149 | 30 (20.1) | 119 (79.9) | 1.5 | 0.8 | 2.6 |
| | Ho Chi Minh | 157 | 46 (29.3) | 111 (70.7) | 2.4 | 1.4 | 4.2 |
| | Can Tho | 152 | 33 (21.7) | 119 (78.3) | 1.6 | 0.9 | 2.9 |
| | Khanh Hoa | 148 | 34 (23) | 114 (77) | 1.7 | 1.0 | 3.1 |
| Sex | Male | 511 | 182 (35.6) | 329 (64.4) | 6.6 | 4.6 | 9.4 |
| | Female | 566 | 44 (7.8) | 522 (92.2) | ref | | |
| Age categories | 18-24 years | 274 | 72 (26.3) | 202 (73.7) | 1.4 | 0.9 | 2.1 |
| | 25-34 years | 266 | 41 (15.4) | 225 (84.6) | 0.7 | 0.5 | 1.1 |
| | 35-44 years | 270 | 59 (21.9) | 211 (78.1) | 1.1 | 0.7 | 1.7 |
| | 45-60 years | 267 | 54 (20.2) | 213 (79.8) | ref | | |
| Marriage | Married | 710 | 134 (18.9) | 576 (81.1) | 0.7 | 0.5 | 0.9 |
| | Single/divorced/widow | 367 | 92 (25.1) | 275 (74.9) | ref | | |
| Education | Elementary | 166 | 36 (21.7) | 130 (78.3) | 1.7 | 1.0 | 2.7 |
| | Junior high school | 259 | 74 (28.6) | 185 (71.4) | 2.4 | 1.6 | 3.7 |
| | High school | 365 | 75 (20.5) | 290 (79.5) | 1.6 | 1.0 | 2.4 |
| | College and higher | 287 | 41 (14.3) | 246 (85.7) | ref | | |
| Occupation | Farmer | 224 | 39 (17.4) | 185 (82.6) | 0.6 | 0.4 | 0.9 |
| | Workers | 93 | 21 (22.6) | 72 (77.4) | 0.9 | 0.5 | 1.5 |
| | Officials | 188 | 24 (12.8) | 164 (87.2) | 0.4 | 0.3 | 0.7 |
| | Others | 572 | 142 (24.8) | 430 (75.2) | ref | | |
| Income categories per month | < 1 million | 221 | 46 (20.8) | 175 (79.2) | 0.6 | 0.3 | 1.0 |
| | 1 - 5 million | 267 | 51 (19.1) | 216 (80.9) | 0.5 | 0.3 | 0.9 |
| | 5-10 million | 508 | 103 (20.3) | 405 (79.7) | 0.5 | 0.3 | 0.9 |
| | >10 million | 81 | 26 (32.1) | 55 (67.9) | ref | | |
| Work requires frequent travel | No | 184 | 26 (14.1) | 158 (85.9) | ref | | |
| | Sometimes | 246 | 44 (17.9) | 202 (82.1) | 1.3 | 0.8 | 2.2 |
| | Often | 647 | 156 (24.1) | 491 (75.9) | 1.9 | 1.2 | 3.0 |
| Drink alcohol | Yes | 361 | 122 (33.8) | 239 (66.2) | 3.0 | 2.2 | 4.1 |
| | No | 716 | 104 (14.5) | 612 (85.5) | ref | | |
| Exercise | No | 599 | 120 (20) | 479 (80) | 0.9 | 0.7 | 1.2 |
| | Yes | 478 | 106 (22.2) | 372 (77.8) | ref | | |

## Optimal cut-off point of CCR to classify active smokers and non-smokers

The CCR is a well-established biomarker for recent nicotine exposure [34–37]. In this study, the optimal cut-off value of 20.947 µg/g creatinine was determined using ROC analysis, which yielded an area under the curve (AUC) of 0.83, indicating good discriminative ability. The sensitivity (70.6%) and specificity (90.1%) at this cut-off demonstrate that the CCR can effectively differentiate smokers from non-smokers in the general adult population.

This cut-off value is consistent with findings from other population-based studies in Asia and Europe, which have proposed CCR thresholds ranging from 15 to 30 µg/g creatinine depending on demographic and environmental factors

[37–41]. However, our estimated threshold (20.947 µg/g) is notably lower than the value reported by Yang et al. (2022), who analysed KNHANES VII data of 3,203 Korean participants aged ≥ 6 years and used a urine cotinine cut-off of 100 µg/g creatinine [42]. It is also lower than the range suggested in Kim's literature review (2019), which reported cut-offs between 33 and 131 µg/g creatinine [43]. These variations across studies may reflect differences in race and ethnicity, smoking behaviour, environmental tobacco smoke exposure, and the implementation of smoke-free policies. In addition, the urinary cotinine cut-off levels were also varied with different measurement methods [44].

Nevertheless, our cut-off value represents a reliable and context-specific biomarker threshold for accurately classifying tobacco exposure in Vietnam, where underreporting of smoking remains a concern. Importantly, CCR offers a key advantage over self-reported data, as it provides an objective measure of exposure unaffected by social desirability bias or recall limitations—issues commonly encountered in tobacco research [14,41,45,46]. This objective measurement is particularly valuable for evaluating the effectiveness of smoke-free policies and for identifying populations at high risk of tobacco exposure.

## Misclassification between self-reported exposure and optimal CCR

Despite the overall correlation between CCR and self-reported exposure, our findings highlight the discordance on smoking status, particularly among smokers. Based on the optimal CCR cut-off, 21.0% of participants were identified as active smokers, higher than the 18.3% who self-reported smoking. This finding is similar to that of previous studies where biomarker results were higher than self-reported results [37,38,41]. This discrepancy suggests an underreporting rate of approximately 2.7%, which is consistent with literature citing the social stigma and reporting bias in smoking behaviours, especially among women or in regions with strong anti-smoking norms [47]. On the other hand, these findings may suggest an excessive environmental exposure to tobacco smoke, usually SHS and THS in Vietnam, especially women who highly expose to tobacco smoke at home and at work [48]. In our study, 19.2% SHS and 3.5% participants from non-smoking households had the CCR level of active smokers. Biomarkers in general and urinary CCR are suitable tools to quantify the tobacco exposure level among SHS and THS which is impossible with self-reported methods.

False positives (CCR ≥ 20.947 µg/g creatinine among self-reported non-smokers) occurred in 9.9% of cases, potentially reflecting misclassified SHS exposure or recent but unreported smoking activity. Conversely, 29.4% of self-reported smokers were misclassified as non-smokers based on CCR < 20.947 µg/g creatinine, possibly due to recent abstinence or interindividual differences in cotinine metabolism [49].

These findings reinforce the importance of incorporating biochemical verification in population-based smoking surveillance. Overreliance on self-report alone may underestimate true exposure and undermine efforts to monitor tobacco control progress.

For instance, the U.S. National Health and Nutrition Examination Survey (NHANES) 2017–2020 reported detectable serum cotinine in approximately 51.2% of non-smoking adults, despite only 22.0% reporting SHS exposure [14]. Similarly, the Korean National Environmental Health Survey (KONEHS) 2015–2017 found that 94.1% of non-smoking adults had detectable urinary cotinine, with geometric mean concentrations of 2.1 µg/L in smoking homes and 1.3 µg/L in smoke-free homes [16]. These findings highlight the limitations of self-reported exposure assessments and emphasise the value of biomarker-based monitoring. Urinary cotinine, with a half-life of approximately 16 hours, serves as a reliable indicator of recent nicotine exposure. The high prevalence of detectable cotinine among non-smokers suggests ongoing involuntary exposure to environmental nicotine sources, including SHS and THS.

## Distribution of smoking status based on the optimal CCR cut-off across groups of study participants

The smoking status which was defined by CCR significantly varied across groups of participants with different sociodemographic and behavioural characteristics. Men had 6.6 times higher odds of being smokers than women, consistent with regional trends and prior research in Southeast Asia [50]. Participants from Ho Chi Minh City and Khanh Hoa also

exhibited significantly higher smoking rates compared to those from Hanoi, suggesting regional variations potentially influenced by urbanisation, lifestyle factors, or enforcement of smoke-free laws.

Education emerged as a protective factor, with participants who completed college or higher education significantly less likely to be smokers. This supports global evidence linking higher education to greater awareness of smoking risks and adherence to health-promoting behaviours [2,51,52]. Similarly, being married was associated with reduced smoking likelihood, possibly reflecting social and familial responsibilities that deter tobacco use.

Income level demonstrated a non-linear association with smoking. Individuals in the highest income group (>10 million VND/month) were more likely to smoke, contrasting with trends in high-income countries where smoking is more prevalent among lower-income populations [2,52]. Our findings are also inconsistent with the previous study in Vietnam, which reported higher smoking prevalence among the lowest income quintile group [53,54]. This may reflect the change in Vietnam's economic context, where rising income can increase access to new tobacco products. Meanwhile, the widespread availability and variety of tobacco products, including affordable and easily accessible e-cigarettes, contribute to the increased prevalence of both active smoking and exposure to second-hand smoke (SHS). This is particularly concerning among young people [55], as e-cigarettes are often marketed with appealing flavours and designs, making them attractive to youth and potentially leading to nicotine addiction and the use of other tobacco products.

Occupational status also played a role, with officials and farmers less likely to smoke compared to freelancers and informal workers. The latter group may experience more stress, irregular routines, or lower exposure to workplace smoking bans, contributing to higher smoking rates [56].

Work-related travel and alcohol consumption were strong behavioural correlates of smoking. Participants who frequently travelled for work had nearly twice the odds of smoking, possibly due to stress, socialisation norms, or reduced oversight. Alcohol drinkers were three times more likely to be smokers—a relationship well-documented in addiction literature due to shared neurobiological pathways and co-use tendencies [57–59].

Demographic analyses reveal higher exposure levels among males, younger adults, and individuals with lower educational attainment, paralleling global evidence linking tobacco exposure to social determinants of health [60–62]. These patterns are consistent with findings from Latin American studies, where vulnerable populations exhibited disproportionate exposure in environments with weak policy enforcement [18]. These demographic disparities highlight the need for targeted public health interventions aimed at reducing nicotine exposure among high-risk groups.

Despite its strengths, including a nationally representative sample, stratified sampling, and standardised biomarker protocols, this study has limitations. First, tobacco exposure was assessed only through a single urinary sample, reflecting short-term exposure (2–4 days). Although urinary cotinine is a robust indicator of recent nicotine exposure, it does not fully capture cumulative or long-term THS exposure. Second, we could not collect the first-void morning urine samples which was ideal for assessing cotinine levels. Due to the design of our community-based study, urine samples were collected onsite only when participants visited the research site. This approach was necessary because we also analysed the samples for metals and other metabolites and therefore could not ensure proper cleanliness and preservation if sampling materials were distributed in advance. To address potential variability in urine dilution, we normalized cotinine results using urinary creatinine concentrations. Third, the absence of environmental measurements (e.g., air nicotine monitors or surface residue) limits our ability to triangulate personal exposure with environmental contamination. Future studies should integrate multi-route assessments, including dust and surface samples, to better quantify THS [9,10].

## Conclusion

Our study highlights the persistent burden of tobacco exposure in Vietnam, the value of urinary CCR as a reliable biomarker, and the misclassification challenges in self-reported data. It also elucidates the sociodemographic and behavioural determinants of smoking, offering valuable insights for tailored tobacco control strategies. Integrating biochemical verification into surveillance systems, alongside targeted interventions for high-risk groups, can enhance the effectiveness of national tobacco control efforts.

## Supporting information

**S1 File. Study questionnaires.**
(XLSX)

**S2 File. PONE-D-25-38585R1 dataset.sav.**
(SAV)

## Acknowledgments

This study used the data and biospecimens collected by the National Project titled "Development of a Basic Biochemical Index Set for Environmental Toxicology in the Vietnamese Working-Age Population" which was implemented by the National Institute of Occupational and Environmental Health. The authors gratefully acknowledge this valuable support. We would like to sincerely thank the project participants.

## Author contributions

**Conceptualization:** Hai Thanh Le, Thuy Thi Thu Tran, Binh Thi Ta, Chinh Thuy Thi Phan, Son Duc Nguyen, Quynh Thuy Nguyen, Bich Ngoc Nguyen, Son Van Nguyen.

**Data curation:** Binh Thi Ta, Huyen Thi Nguyen, Van Thi Pham, Chinh Thuy Thi Phan, Nhat Minh Tu.

**Formal analysis:** Thuy Thi Thu Tran, Binh Thi Ta, Chinh Thuy Thi Phan.

**Funding acquisition:** Hai Thanh Le, Son Duc Nguyen, Son Van Nguyen.

**Investigation:** Binh Thi Ta, Huyen Thi Nguyen, Van Thi Pham, Chinh Thuy Thi Phan, Nhat Minh Tu.

**Methodology:** Thuy Thi Thu Tran, Binh Thi Ta, Huyen Thi Nguyen, Van Thi Pham, Son Duc Nguyen, Quynh Thuy Nguyen, Bich Ngoc Nguyen, Son Van Nguyen.

**Project administration:** Binh Thi Ta, Huyen Thi Nguyen, Van Thi Pham, Chinh Thuy Thi Phan, Nhat Minh Tu.

**Resources:** Hai Thanh Le, Binh Thi Ta, Huyen Thi Nguyen, Van Thi Pham, Nhat Minh Tu.

**Supervision:** Hai Thanh Le, Son Duc Nguyen, Quynh Thuy Nguyen, Bich Ngoc Nguyen, Son Van Nguyen.

**Validation:** Thuy Thi Thu Tran.

**Visualization:** Thuy Thi Thu Tran.

**Writing – original draft:** Thuy Thi Thu Tran, Binh Thi Ta, Chinh Thuy Thi Phan.

**Writing – review & editing:** Hai Thanh Le, Thuy Thi Thu Tran, Binh Thi Ta, Chinh Thuy Thi Phan, Son Duc Nguyen, Quynh Thuy Nguyen, Bich Ngoc Nguyen, Son Van Nguyen.

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
