## [Decision Letter · Decision Letter 0]

29 Sep 2025

Dear Dr. Tran,

We look forward to receiving your revised manuscript.

Kind regards,

Iman Al-Saleh

Academic Editor

PLOS ONE

Journal Requirements:

“This study was supported by the National Project “Development of a Basic Biochemical Index Set for Environmental Toxicology in the Vietnamese Working-Age Population” (Project code: ĐTĐL.CN-39/21), under the Basic Science Development Program in Chemistry, Life Sciences, Earth Sciences, and Marine Sciences (2017–2025) funded by the Ministry of Science and Technology. The authors gratefully acknowledge this valuable support”

“SVN received the funding from the Ministry of Science and Technology as a principle investigator of the National Project “Development of a Basic Biochemical Index Set for Environmental Toxicology in the Vietnamese Working-Age Population” (Project code: ĐTĐL.CN-39/21) under the Basic Science Development Program in Chemistry, Life Sciences, Earth Sciences, and Marine Sciences (2017–2025). The funders had no role in study design, data collection and analysis, decision to publish, or preparation of the manuscript.”

Reviewers' comments:

Reviewer's Responses to Questions

**Comments to the Author**

1. Is the manuscript technically sound, and do the data support the conclusions?

Reviewer #1: Yes

Reviewer #2: Yes

Reviewer #3: Partly

2. Has the statistical analysis been performed appropriately and rigorously?

Reviewer #1: Yes

Reviewer #2: Yes

Reviewer #3: Yes

3. Have the authors made all data underlying the findings in their manuscript fully available?

Reviewer #1: Yes

Reviewer #2: Yes

Reviewer #3: Yes

4. Is the manuscript presented in an intelligible fashion and written in standard English?

Reviewer #1: Yes

Reviewer #2: Yes

Reviewer #3: No

**Reviewer #1**: This study presents a nationwide analysis of urinary cotinine as a biomarker for exposure to tobacco smoke (nicotine). The study itself is not novel. Surveys on tobacco smoke exposure are routinely conducted in various countries, including the USA, Brazil, Korea, and Japan. However, the introduction of a cut-point for urinary cotinine levels, adjusted for creatinine concentration, adds valuable insight to the literature.

Overall, the study is well-executed, with data processed through robust statistical analysis, yielding reliable results. Here are some suggestions for enhancing this work:

1. The urinary cotinine concentrations for active and passive smokers in this study should be compared with those in other global studies, and any differences should be discussed. Please review additional literature on this topic. The reviewer can suggest a few as follows, sorted by publication year:

- 2022: https://doi.org/10.3390/ijerph19063746

- 2021: https://doi.org/10.1186/s12889-021-11265-y

- 2021: https://doi.org/10.4082/kjfm.20.0056

- 2020: http://dx.doi.org/10.2174/1875318302010010060

- 2020: https://doi.org/10.21215/kjfp.2020.10.5.378

- 2019: https://doi.org/10.1080/1354750X.2019.1684563

- 2019: https://doi.org/10.1080/1354750X.2020.1797879

2. Similarly, the proposed cut-point should be compared with those suggested in other studies. In several studies, the cut-point for urinary cotinine is suggested to range from 30 to 40 µg/g-creatinine. Please review additional literature on this topic. The reviewer can suggest a few as follows, sorted by publication year:

- 2022: https://doi.org/10.3390/ijerph19063746

- 2020: https://doi.org/10.3390/ijerph17155537

- 2016: https://doi.org/10.3390/ijerph13121236

3. Surprisingly, the survey does not inquire about the types of cigarettes regularly smoked by participants. With the increasing prevalence of vaping, there may be greater nicotine exposure compared to conventional smoking. This is a significant limitation and should be discussed comprehensively.

4. Please provide QA/QC information for the analysis.

**Reviewer #2:**Dear Authors,

Thank you for submitting your manuscript to PLOS ONE. In your statistical analysis (lines 215, 216, and 217), please specify the exact outlier detection test you used. For example, did you use the Grubbs test (https://www.graphpad.com/quickcalcs/) for continuous data? If so, please indicate this clearly and, if desired, provide the reference. At the end of line 236, ensure you include the confidence interval used (e.g., 95%), as this information is essential. For Figure 1, please indicate the sensitivity and specificity test values, alongside the CCR cut-off value, within the figure or its legend, as the cut-off selection in the ROC curve depends on these metrics.

I hope these minor suggestions will assist you in improving the manuscript.

**Reviewer #3**: This manuscript describes the first nation-wide study to monitor smoking prevalence and exposure to second hand smoke in Vietnam. So far, most studies in Vietnam and developing countries only used questionnaires to record and estimate prevalence, thus the data are of great interest to public health agencies and could serve as evidence for future tobacco control policies.

The authors have presented all the data, with unexpected findings such as lower smoking prevalence in farmers and lower income earners than official and high income earners.

However, I suggest the authors to consider some issues below:

1. The recruitment process has to be described in detail to show that the sampled populations are representative.

2. The justification of spot urine is not correct in my opinion. First void samples will probably give more consistency and be complemented by the questionnaires about how many cigarettes you smoked yesterday

3. I am not an expert in statistics but I wonder whether the unexpected findings such as the low prevalence in farmers and high prevalence in Ho Chi Minh city have anything to do with the power of the data (i.e. the sample size at provincial level is small and thus prone to high uncertainty).

4. The data about female smoking prevalence is also interesting as people usually assume very low level of smoking in female in Asia in general. Question to the authors is that whether the relatively high prevalence in female is due to the “false positive” diagnosis using the CCR of 21 (I don’t think you need to be super accurate with 3 significant figures).

5. Similarly, the discussion about under-reporting of smoking status, my layman understanding is that the research team, not only of this study but all studies, would have had to declare in the Participant Information Sheet that they would analyse the urine samples for tobacco biomarkers. Therefore, it would be a real surprise if the participants still tried to hide their smoking status in the questionnaires. To me, it may well be the effect of the artificial CCR values.

6. If possible, try to present the data in figures rather than table form.

7. If possible, please compile a range of CCR values from different studies.

**Do you want your identity to be public for this peer review?** For information about this choice, including consent withdrawal, please see our Privacy Policy

Reviewer #1: No

Reviewer #2: **Yes:** Ricardo David Couto, Department of Clinical and Toxicological Analysis, Faculty of Pharmacy, Federal University of Bahia, Brazil.

Reviewer #3: No

---

## [Author Response · Author response to Decision Letter 1]

8 Nov 2025

PONE-D-25-38585

Urinary Cotinine concentration as a biomarker of environmental exposure to Nicotine in Vietnam: Results from a Nationwide Survey in 2024

Journal Requirements:

Reply:

The authors prepared the revised manuscript following PLOS ONE's style requirements/ templates.

“This study was supported by the National Project “Development of a Basic Biochemical Index Set for Environmental Toxicology in the Vietnamese Working-Age Population” (Project code: ĐTĐL.CN-39/21), under the Basic Science Development Program in Chemistry, Life Sciences, Earth Sciences, and Marine Sciences (2017–2025) funded by the Ministry of Science and Technology. The authors gratefully acknowledge this valuable support”

“SVN received the funding from the Ministry of Science and Technology as a principle investigator of the National Project “Development of a Basic Biochemical Index Set for Environmental Toxicology in the Vietnamese Working-Age Population” (Project code: ĐTĐL.CN-39/21) under the Basic Science Development Program in Chemistry, Life Sciences, Earth Sciences, and Marine Sciences (2017–2025). The funders had no role in study design, data collection and analysis, decision to publish, or preparation of the manuscript.”

Reply:

The authors revised the Acknowledgments Section (line 508-512) as follows:

“This study used the data and biospecimens collected by the National Project titled “Development of a Basic Biochemical Index Set for Environmental Toxicology in the Vietnamese Working-Age Population” which was implemented by the National Institute of Occupational and Environmental Health. The authors gratefully acknowledge this valuable support. We would like to sincerely thank the project participants”.

The authors would like to udpate the Funding Statement as follows:

“The National Project “Development of a Basic Biochemical Index Set for Environmental Toxicology in the Vietnamese Working-Age Population” (Project code: DTDL.CN-39/21) was funded by the Ministry of Science and Technology, Vietnam, under the Basic Science Development Program in Chemistry, Life Sciences, Earth Sciences, and Marine Sciences (2017–2025). The funders had no role in study design, data collection and analysis, decision to publish, or preparation of this manuscript.”

Reply:

Thank you very much for your guidance, the authors reviewed and selected appropriate references with thorough consideration.

Reply to Reviewer: 1

This study presents a nationwide analysis of urinary cotinine as a biomarker for exposure to tobacco smoke (nicotine). The study itself is not novel. Surveys on tobacco smoke exposure are routinely conducted in various countries, including the USA, Brazil, Korea, and Japan. However, the introduction of a cut-point for urinary cotinine levels, adjusted for creatinine concentration, adds valuable insight to the literature.

Overall, the study is well-executed, with data processed through robust statistical analysis, yielding reliable results. Here are some suggestions for enhancing this work:

#1. The urinary cotinine concentrations for active and passive smokers in this study should be compared with those in other global studies, and any differences should be discussed. Please review additional literature on this topic. The reviewer can suggest a few as follows, sorted by publication year:

- 2022: https://doi.org/10.3390/ijerph19063746

- 2021: https://doi.org/10.1186/s12889-021-11265-y

- 2021: https://doi.org/10.4082/kjfm.20.0056

- 2020: http://dx.doi.org/10.2174/1875318302010010060

- 2020: https://doi.org/10.21215/kjfp.2020.10.5.378

- 2019: https://doi.org/10.1080/1354750X.2019.1684563

- 2019: https://doi.org/10.1080/1354750X.2020.1797879

Reply

The authors highly appreciated your surport. We thoroughly reviewed all suggested references. However, we will provide further discussion with the results of Yang et al (2022) (reference https://doi.org/10.3390/ijerph19063746: Exposure and Risk Assessment of Second- and Third-Hand Tobacco Smoke Using Urinary Cotinine Levels in South Korea(1)) which analyzed KNHANES VII data on 3203 Korean subjects (aged over 6 years old) during the period 2016–2018. Yang’s study used the urine cotinine cutoff of 100 μg/g-creatinine. Another reference is from Kim et al, (2019), in their literature review, the authors suggested the urine cotinine cutoffs will be approximately 33–131 μg/g-creatinine (2).

Other studies used the geometric mean of urinary cotinine concentrations or urinary cotinine levels that were different from the depedent variables of our study which were urinary cotinine to creatinine concentration/ cutoff levels. This difference makes comparison quite inappropriate.

Additional discussion was provided from line 386 to 404 under section Discussion � Optimal Cut-off Point of CCR to Classify Active Smokers and Non-Smokers.

#2. 2. Similarly, the proposed cut-point should be compared with those suggested in other studies. In several studies, the cut-point for urinary cotinine is suggested to range from 30 to 40 µg/g-creatinine. Please review additional literature on this topic. The reviewer can suggest a few as follows, sorted by publication year:

- 2022: https://doi.org/10.3390/ijerph19063746

- 2020: https://doi.org/10.3390/ijerph17155537

- 2016: https://doi.org/10.3390/ijerph13121236

Reply

The authors highly appreciated your surport and valuable suggestions. We thoroughly reviewed all suggested references.

We will add further appropriate discussions with the results of Yang et al (2022) (https://doi.org/10.3390/ijerph19063746) as response to the previous comments.We also add further appropriate dicussion with results with Kim et al (2016) (https://doi.org/10.3390/ijerph13121236) which discussed that the urinary cotinine cutoff levels were varied with different measurement methods (3). These discussions were provided from line 386 to 404 under section Discussion � Optimal Cut-off Point of CCR to Classify Active Smokers and Non-Smokers

We do not have further discussion with the results of Nishihama et al (2020) (https://doi.org/10.3390/ijerph17155537) because this study was conducted with 89,895 pregnant women who participated in the Japan Environment and Children’s Study in September 2019. In our study, we only included healthy adults who were not in their pregnancy so comparison of the two studies’ results might not be appropriate.

#3. Surprisingly, the survey does not inquire about the types of cigarettes regularly smoked by participants. With the increasing prevalence of vaping, there may be greater nicotine exposure compared to conventional smoking. This is a significant limitation and should be discussed comprehensively.

Reply

Thank you for your comment. In this study, our primary objective was to quantify tobacco smoke exposure among adults in Vietnam in 2024 using urinary cotinine to creatinine concentrations. Since nicotine from all tobacco products is metabolized into cotinine, our assessment focused on cotinine levels regardless of the specific type of tobacco exposure. We collected self-reported information on cigarette and pipe tobacco use; however, we did not distinguish between different forms of tobacco products (e.g., paper cigarettes, electronic cigarettes, etc.) in our analysis.

#4. Please provide QA/QC information for the analysis

Reply

Thank you for pointing this out. We providied additional QA/QC information in the Laboratory Analysis section (line 230-237) as follow:

Certified reference materials (CRM) ClinChek® Urine Controls, Germany level I were used for quality control the results of LC MS/MS analysis. Measured results and recoveries for cotinine in CRM ClinChek® Urine Controls, Germany are shown in the table 1. CRM has a target value of 163 µg/L, the allowable value range is from 130 µg/L to 196 µg/L.

Table 1. Measured results and recoveries for cotinine in Certified reference materials ClinChek® Urine Controls

No Sample Cotinine concentration (µg/L) Recovery

(%) Average cotinine concentration (µg/L) Standard deviation (SD) Relative standard deviation (RSD%)

1 CRM1 162.69 99.81 166.16 7.74 4.66

2 CRM2 165.39 101.47

3 CRM3 173.47 106.43

4 CRM4 170.78 104.77

5 CRM5 176.17 108.08

6 CRM6 160.00 98.16

7 CRM7 154.61 94.85

Reply to reviewer 2:

#1. In your statistical analysis (lines 215, 216, and 217), please specify the exact outlier detection test you used. For example, did you use the Grubbs test (https://www.graphpad.com/quickcalcs/) for continuous data? If so, please indicate this clearly and, if desired, provide the reference.

Reply

Thank you for your constructive comment regarding data variability. To ensure robust interpretation of cotinine-creatinine corrected concentrations, we conducted a comprehensive outlier assessment using multiple statistical approaches (IQR rule, z-score threshold, and percentile method), along with a biological plausibility review. Values flagged by at least two methods were classified as outliers (32 participants; 2.97%). Instead of excluding them directly, we retained these observations but marked them with an indicator variable to support sensitivity analyses and avoid bias introduced by data removal. Additional explanation was provided in the Statistical Analysis (line 255-262) as follow:

“Cotinine concentrations normalized by urinary creatinine were evaluated for extreme values using multiple statistical approaches, including the interquartile range (IQR) rule, z-scores (> |3|), and the 99th percentile threshold. A biological plausibility check was also performed based on published reference ranges for urinary cotinine in smokers. Data points that were identified as extreme by at least two independent methods were classified as outliers. Using this combined approach, 32 outliers (2.97% of total samples) were flagged. These values were retained in the dataset with an indicator variable for sensitivity analyses and model robustness checks.”

#2. At the end of line 236, ensure you include the confidence interval used (e.g., 95%), as this information is essential.

Reply

Thank you for pointing this out. The authors indicated the use of odd ratios and 95% confidence interval in the Statistical Analysis section line 276.

#3. For Figure 1, please indicate the sensitivity and specificity test values, alongside the CCR cut-off value, within the figure or its legend, as the cut-off selection in the ROC curve depends on these metrics.

Reply

Thank you for your comments, the authors included the sensitivity and specificity test values, CCR cut-off values and AUC in the Figure 1 in line 311.

Reply to reviewer 3:

This manuscript describes the first nation-wide study to monitor smoking prevalence and exposure to second hand smoke in Vietnam. So far, most studies in Vietnam and developing countries only used questionnaires to record and estimate prevalence, thus the data are of great interest to public health agencies and could serve as evidence for future tobacco control policies.

The authors have presented all the data, with unexpected findings such as lower smoking prevalence in farmers and lower income earners than official and high income earners.

However, I suggest the authors to consider some issues below:

#1. The recruitment process has to be described in detail to show that the sampled populations are representative.

Reply

Thank you for pointing this out. The authors provided details on the sampling method in the Study Design and Population section line 153 to 176 as follow:

“Study participants were selected via multi-stage sampling procedure as follow:

Stage 1. Selecting provinces: To ensure that the research sample represents the Vietnamese working-age population in different parts of Vietnam, we randomly selected one province from each of seven administrative and socio-ecological regions of the country. The selected provinces included Lang Son province (Northeast region), Phu Tho province (old Hoa Binh, Northwest region), Hanoi city (Red River Delta region), Nghe An province (North Central region), Khanh Hoa province (South Central Coast and Central Highlands region). Ho Chi Minh city (Southeast region), Can Tho city (Mekong Delta region).

Stage 2. Selecting commune/ward areas: In each province/city, we randomly selected one district/county. In each district/county, we intentionally selected one ward/commune that were not located near mining, metallurgy, pesticide production, large industrial zones, had no craft villages with the risk of exposure to toxic chemicals, and no cultivation of industrial crops. There were seven ward/communes were invited to join the study.

Phase 3. Selecting subjects. Local health workers under the guidance of study staff prepared a list of subjects who met the study inclusion criteria. Subjects were stratified by sex (approximately 1:1 male-to-female ratio) and across four age groups (18–24, 25–34, 35–44, 45–60 years). From the above list, 20 subjects were randomly selected in each age group with equal sex ratio. Thus, 160 subjects (80 men and 80 women) were invited to join the study in each commune/ward. During the data collection process, if the subject refused to join or end their participation, another study subject would be randomly selected from the initial list. In total, 1077 healthy Vietnamese people in working age participated in the study”

#2. The justification of spot urine is not correct in my opinion. First void samples will probably give more consistency and be complemented by the questionnaires about how many cigarettes you smoked yesterday

Reply

Thank you for your valuable comments. We agree that collecting first-void morning urine samples is ideal for assessing cotinine levels. However, due to the design of our community-based study, urine samples were collected onsite only when participants visited the research site. This approach was necessary because we also analyzed the samples for metals and other metabolites, and therefore could not ensure proper cleanliness and preservation if sampling materials were distributed in advance. To address potential variability in urine dilution, we normalized cotinine results using urinary creatinine concentrations. We also include this as one of other limitations of our study at the end of the discusison section (line 488-495)

#3. I am not an expert in statistics but I wonder whether the unexpected findings such as the low prevalence in farmers and high prevalence in Ho Chi Minh city have anything to do with the power of the data (i.e. the sample size at provincial level is small and thus prone to high uncertainty).

Reply

The unexpectedly low prevalence among farmers and the higher prevalence in Ho Chi Minh City may be explained by geographical variability in smoking behavior, as well as differences in accessibility and affordability of tobacco products. For example, despite being the second larges

---

## [Decision Letter · Decision Letter 1]

23 Nov 2025

Urinary Cotinine concentration as a biomarker of environmental exposure to Nicotine in Vietnam: Results from a Nationwide Survey in 2024

PONE-D-25-38585R1

Dear Dr. Tran,

We’re pleased to inform you that your manuscript has been judged scientifically suitable for publication and will be formally accepted for publication once it meets all outstanding technical requirements.

Kind regards,

Iman Al-Saleh

Academic Editor

PLOS ONE

Additional Editor Comments (optional):

Reviewers' comments:

Reviewer's Responses to Questions

**Comments to the Author**

Reviewer #1: All comments have been addressed

Reviewer #2: All comments have been addressed

Reviewer #3: All comments have been addressed

2. Is the manuscript technically sound, and do the data support the conclusions?

Reviewer #1: Yes

Reviewer #2: Yes

Reviewer #3: Yes

3. Has the statistical analysis been performed appropriately and rigorously?

Reviewer #1: Yes

Reviewer #2: Yes

Reviewer #3: (No Response)

4. Have the authors made all data underlying the findings in their manuscript fully available?

Reviewer #1: Yes

Reviewer #2: Yes

Reviewer #3: Yes

5. Is the manuscript presented in an intelligible fashion and written in standard English?

Reviewer #1: Yes

Reviewer #2: Yes

Reviewer #3: Yes

Reviewer #1: The presented responses were thoughtful, and the manuscript is well-revised.

The reviewer has no further comments.

Congratulations to the authors for running such a meaningful survey.

Reviewer #2: (No Response)

Reviewer #3: the authors have addressed the comments adequately although I feel the title and the conclusion are not well aligned, please consider revise the title.

**Do you want your identity to be public for this peer review?** For information about this choice, including consent withdrawal, please see our Privacy Policy

Reviewer #1: No

Reviewer #2: **Yes:** Ricardo David Couto

Reviewer #3: No

---

## [Editor Report · Acceptance letter]

PONE-D-25-38585R1

PLOS One

Dear Dr. Tran,

I'm pleased to inform you that your manuscript has been deemed suitable for publication in PLOS One. Congratulations! Your manuscript is now being handed over to our production team.

Kind regards,

on behalf of

Dr. Iman Al-Saleh

Academic Editor

PLOS One